# Memorable Tourism Experiences' Formation Mechanism in Cultural Creative Tourism: From the Perspective of Embodied Cognition

**Ziyan Yin** *, **Anmin Huang and Jiashu Wang**

College of Tourism, Huaqiao University, Quanzhou 362021, China
* Correspondence: 17013121019@stu.hqu.edu.cn

**Abstract:** Drawing on embodied cognition theory, this study treated embodied engagement as an antecedent to memorable tourism experiences. Grounded theory and content analysis methods were used to construct the formation mechanism model of memorable tourism experiences in cultural creative tourism. The concept and characteristics of memorable tourism experiences were also discussed. The results indicated that embodied and thinking engagements had positive effects on cognitive evaluation and emotion, which further led to memorable tourism experiences. Visitor interaction played a regulating role in the influence of embodied and thinking engagements on cognitive evaluation and emotion. This study provided important implications for tourism research and practice.

**Keywords:** memorable tourism experiences; formation mechanism; embodied cognition theory; grounded theory; cultural creative tourism

## 1. Introduction

When consumers are faced with the choice of travel destination again, the memory of previous travel experience will be recalled unconsciously, and people will make decisions by memory [1,2]. Although most scholars agree that satisfaction is the main index to predict tourists' revisit intentions [3,4], some scholars believe that, compared with satisfaction, memory (as a sublimation of tourism experience) is a more accurate indicator [5–7]. Tourists tend to revisit destinations that bring them positive memories [8]. For example, Chen and Rahman, taking cross-cultural interaction as the background, proposed that memorable tourism experiences would affect tourists' loyalty and intention of revisiting destinations [9]. Determining how to help travelers to gain memorable experiences is a crucial goal for tourism product development and efficient resource allocation [6]. Memorable tourism experiences could affect tourists' behavior intentions and tourism destination competitiveness. As such, it has become a hot topic in academia. However, Sthapit and Coudounaris (2018) emphasized that memorable tourism experiences would vary from case to case [10]. This was also noted by Seyfi, Hall, and Rasoolimanesh (2020), who called for MTEs studies to be more contextualized [11]. The field of research on MTEs needs to be further expanded. This will be conducive to further exploration of the uniqueness of memorable tourism experiences in different contexts.

The existing studies emphasized the importance of memorable tourism experiences by exploring the relationship between memorable tourism experiences and behavior intention [12–14], as well as place attachment [15]. Although some studies analyzed the antecedents of memorable tourism experiences, the decisive factors and mechanisms have not yet been made clear. Scholars have explored the positive effects of pleasure, novelty, engagement, and emotional change on memorable tourism experiences [6,16–18]. Kim (2012) attempted to develop the seven-factor worth-remembering experience scale and confirmed its content validity and reliability in subsequent studies [19]. However, most

studies only focused on the elements of memorable on-site experiences; explorations of the formation mechanism of memorable tourism experiences have been slight.

Cultural creative tourism is a more consolidated form of tourism that integrates the cultural creative industries. It has great vitality under the background of the experience economy [20]. Cultural creative tourism takes culture and creativity as core attractions and emphasizes the active participation of tourists, which provides more possibilities for tourists to create memorable travel experiences [21–24]. These memories influence the formation of tourists' long-term attitudes toward the destination [25]. Therefore, attracting more potential tourists and establishing a positive tourism image by managing tourists' experiential memories is profound. Further empirical analysis of the memorable tourism experiences of typical tourism destinations are required, in the context of cultural tourism integration.

What are the factors that affect memorable tourism experiences in cultural creative tourism, and how do these factors interact? The embodied cognition theory emphasizes that the tourism experience is produced by bodily experience. Tourism experience quality is a map of the psychological and physiological satisfaction of tourists, but it is also dependent upon the situation [26]. This embodiment has gradually become an important perspective of the tourism experience and has even come to represent a new paradigm of tourism experience research [27–30]. Cultural creative tourists are active and curious; physical participation and involvement is an important way to optimize cultural creative experiences [31]. Previous studies have shown the positive impact of sensory experiences on memorable tourism experiences [13]. Therefore, taking embodied experience as a breakthrough, from a new perspective, will prove helpful to clarify the influencing factors and forming mechanisms of the memorable tourism experience in cultural creative tourism.

Based on the above background, this study adopted grounded theory and content analysis methods to analyze data collected through in-depth interviews and user-generated content obtained using crawling software. This study aimed to achieve the following objectives: (1) extraction of the key factors determining memorable tourism experiences, and establishment of new components of memorable tourism experiences based on cultural creative tourism; (2) from the perspective of embodied cognition theory, construction of a formation mechanism model of memorable tourism experiences in cultural creative tourism; (3) further exploration of the concept and characteristics of memorable tourism experiences. The key point of this study was to describe the tourism experience from a memory-based perspective. The MTEs theory was added to the context of cultural creative tourism to improve the general tourism experience model. This study will help to further understand the essential laws of memorable tourism experiences. Additionally, a new practical way was established to enhance tourists' positive tourism memory and reduce their negative tourism memories, which will be conducive to increasing revisit rates.

## 2. Literature Review

### 2.1. Cultural Creative Tourism

With deep historical and cultural heritage, cultural creative tourism integrates new creative elements to create a popular, forward-looking, healthy life experience. As such, it has become a new bright spot to attract tourists. At present, the concept of cultural creative tourism is confused. The academic circle has put forward several similar concepts about cultural creative tourism, such as cultural and creative tourism, creative cultural tourism, and creative tourism. Scholars define it from different perspectives, such as tourists, industry integration, product development, and tourism development mode. Cultural and creative tourism emphasizes the integration of cultural and creative industries. Creative cultural tourism is a new mode of cultural tourism [32]. Cultural creative tourism is based on the foundation of creative tourism and emphasizes the importance of culture—especially traditional Chinese culture, in this study—which belongs to the category of creative tourism. The roots of creative tourism go back to the mid-1990s, when a group of researchers and practitioners were looking at ways to enhance the sales of craft products to tourists ( . . . )

through the project EUROTEX [33,34]. Pearce and Butler (1993) first identified creative tourism as a potential form of tourism [34]. Richards and Raymond (2000) first proposed creative tourism as a form of tourism in which tourists could stimulate their creative potential and further experience the cultural atmosphere by actively participating in the cultural or technical learning of the destination country or community [31]. Other scholars have further elaborated on the connotation of cultural creative tourism and pointed out that culture is the core of cultural creative tourism, and creativity is a means of cultural creative tourism [21–24]. As a form of post-modern tourism consumption, its concept has evolved throughout the years [34,35]. In general, however, the definition of cultural creative tourism must focus on: the development of the creative potential of tourists; the active involvement of the tourist; personal experiences of the chosen destination; and co-creation as a process of production of products, services, and experiences [36]. Cultural creative tourism presents the following two characteristics: on the one hand, cultural creative products are the core attraction of cultural creative tourism. Culture is the foundation, and creativity is the means. Unique culture and creativity are easily impressive and create memorable tourism experiences [37,38]. On the other hand, cultural creative tourism emphasizes tourist participation and is committed to developing the creative potential of visitors through co-creation. Due to the immateriality of culture and creativity, tourists' proactiveness is vital to cultural creative tourism destinations. In addition, cultural creative tourists have higher demands and higher standards for tourism products. Gradually, such tourists will grow unsatisfied with the passive absorption of culture, instead actively seeking positive cultural experience activities [39,40]. Therefore, tourist participation is one of the most important characteristics of cultural creative tourism [31], and is crucial to the formation of memorable tourism experiences. It is typical and representative to study memorable tourism experiences in cultural creative tourism. Some scholars have also discussed the relationships between memory, creative experiences, and satisfaction [41]. They found that memory played a crucial role in predicting the willingness to revisit in the context of cultural creative tourism. However, research on the antecedent variables of memorable tourism experiences and the formation mechanism in cultural creative tourism could be further discussed.

*2.2. Memorable Tourism Experiences*

Unexpected, emotional, and important experiences can create vivid and lasting memories in the individual mind [42]. Tourism experience is different from the daily life experience, the emotional accumulation of individual tourists can impress visitors [43]. Memorable tourism experiences are defined as positive remembered tourism experiences after an event [19]. They represent autobiographical memory, because they are closely connected to tourist participation and self-emotion [19]. The essence of memorable tourism experiences is storytelling [6,19].

Kim et al. (2012) developed the memorable tourism experiences scale. They summarized seven influencing variables: happiness, rest, social interaction, cultural factors, freshness, involvement, unique meaning, and knowledge [19]. This statement was comprehensive and has been highly recognized in academia. Before and after Kim et al.'s (2012) study, other scholars also conducted specific and in-depth research on the above aspects [44,45]. Chandralal and Rindfleish (2015) found that blogs mainly reported positive experience memories and memorable tourism experiences, including the seven themes of local peoples' life and culture, personal important experiences, experiences sharing, novelty, accidents, tour guide and travel agency services, and related emotions [46].

The factors that affect memorable tourism experiences can be divided into internal and external factors [47]. Tourists' motivation, past experiences, understanding of the destination, and personal character are internal factors, while external factors include the induced marketing images relating to the destination, travel activities, patterns of change at the place, and the people with whom the destination is shared [48]. In addition, many studies have been devoted to investigating the impact of memorable tourism experi-

ences on tourist behavioral intention, such as willingness to revisit, satisfaction, word of mouth, etc. [9,49–51]. Tsai (2016) conducted a study based on tourists' food experiences and found that memorable tourism experiences would affect tourists' loyalty [13].

In summary, research on memorable tourism experiences involve the concepts, structure dimensions, variable measurements, influence factors, and influences on behavior intention, etc., but most of the research about memorable tourism experiences only involves the on-site factors. However, memory will continue to develop and change [52]. The formation and change process of memorable tourism experiences should be further discussed. This paper aimed to answer these key questions and make up for the shortcomings of previous studies. In addition, the connotations of memorable tourism experiences change as situations change [10]. The research context of memorable travel experiences includes specific destinations in the general tourism context, food and wine tourism, and non-general tourism contexts, such as ethnic minorities, heritage, cultural tourism, nature tourism, smart tourism, educational holidays, coffee farms, festivals, sports tourism, hotels, Airbnbs, etc. [53]. However, most studies have applied the maturity scale to different tourism situations, making it difficult to find the uniqueness of memorable tourism experiences in various tourism situations. In the field of cultural tourism, Seyfi et al. established a theoretical model of memorable cultural tourism experiences and proposed key influencing factors of the experience, such as the prior perceived significance of the experience, authenticity, engagement, cultural exchange, culinary attraction, and quality of service [11]. That provided a reference for this study, but there were still deficiencies in the discussion of the relationship between each factor. Additionally, they ignored the excellent function of creative elements in memorable tourism experiences, in the context of the dynamic development of tourism. Therefore, the uniqueness of memorable tourism experiences and the formation mechanisms of cultural creative tourism needed to be further clarified in the context of cultural and tourism integration.

### 2.3. Embodied Cognitive Theory

In Phenomenology of Perception, Merleau Ponty regarded the body as the subject of our perceptual world. The embodied cognition theory holds that human cognition is deeply rooted in the interaction between the body and the world. The embodied participation through the contact of body organs and landscape resources to achieve the purpose of experience [54]. In recent years, some scholars have proposed embodiment as the main attribute of the tourism experience [30,55,56]. The embodied experience began to reflect on the absence of the body in previous tourism experience research. An increasing number of scholars began to emphasize the importance of other senses of the body, such as a tourist experience at a music festival being affected by hearing [57], a tourist experience on the beach being affected by touch, and a tourist experience of street food being affected by taste [26,58]. The conceptual framework of embodied experience emphasizes the interaction among body, perception, and situation, all of which affect tourists' cognition of the local meaning [29,59]. Memory, as the sublimation of experience, is closely related to tourists' embodied participation, environment perception, and meaning cognition. Cultural creative tourism takes culture and creativity as its core and has distinct creative expression and visual effects [60]. The rich and colorful cultural activities and attractive external environment are conducive to stimulating the enthusiasm of tourists to participate in embodied tourism activities [61], thus providing good conditions for creating memorable tourism experiences. Therefore, embodied cognition theory may be an important perspective for the study of cultural creative tourism memory experiences.

## 3. Methods

### 3.1. Grounded Theory and Content Analysis Method

The grounded theory was put forward by Glaser and Strauss (1968), aimed at theory construction [62]. The grounded theory seeks the core concept reflecting social phenomena and the relationship among concepts through coding. It can establish substantive theory

from bottom to top [63]. It centers on discovery logic rather than validation logic. Coding is divided into open coding, axial coding, and selective coding, which is the process of conceptualizing, categorizing, and theorizing the original data. Compared with quantitative research methods, coding is more suitable for combining and studying complex text materials; it is often used to explore and refine research constructs and composition dimensions, and it is also useful in analyzing influencing factors and interaction relationships [64,65]. Therefore, this article used NVivo software to analyze the text data according to the analysis steps and framework of grounded theory. This was helpful in efforts to identify what constituted memorable tourism experiences and construct its formation mechanism from bottom to top in cultural creative tourism.

The content analysis combined quantitative and qualitative methods. It was able to evaluate the text data collected in various ways objectively, systematically, and quantitatively [66]. To make up for the strong subjectivity of the interview method, tourism comments and travel notes on memorable tourism experiences in cultural creative tourism were collected through the network. After coding and classifying, we used ROST content mining software to analyze high-frequency feature words for detecting the saturation and interpretability of the constructed model.

### 3.2. Sample and Data Collection

Given the research background, this study selected three cultural creative destinations as research cases: Wudianshi, Jinjiang City, Fujian Province; Kuanzhai Alley, Chengdu City, Sichuan Province; and Nanluoguxiang, Beijing City. Wudianshi traditional street has been identified as a national night culture and tourism consumption gathering area. Kuanzhai Alley won the title of "Sichuan Historical and Cultural Street". Nanluoguxiang is one of the oldest neighborhoods in Beijing, and it completely preserves the courtyard texture of the Yuan Dynasty in China; it is also rich in the Beijing style. These were chosen for three reasons. First, the three cases were destinations based on unique culture, with many creative elements, emphasizing the active participation of tourists. Second, the three cultural creative destinations covered different regions of China. Last, they were all popular cultural creative tourism destinations, so it was easier to conduct interviews and online text collections.

The researchers determined the interview outline design after consulting and comparing authoritative literature and combining the opinions of two masters, engaged in tourism management research, and two professors, engaged in cultural creative tourism research. As shown in Appendix A, the outline of the interview focused on the experience and memory of tourists in cultural creative tourism, including questions such as:

- "Why do you choose to travel to this cultural creative tourism destination?"
- "What impressed you most during your visit to the cultural creative tourism destination?"
- "What were your favorite and least favorite aspects of the tour?"
- "How has your evaluation and attitude of that trip changed compared to the onsite experiences?"
- "What factors will you consider when you choose to travel to a cultural creative tourism destination again?"

There were, in total, 9 questions, in addition to the basic information of respondents, such as gender, age, and occupation.

This paper adopted the methods of convenience sampling, intention sampling, and snowball sampling to invite qualified relatives, friends, classmates, hornet's nest tourists, field tourists, and tourists recommended as interviewees. The selection of interviewees followed three criteria:(1) tourists who have visited Jinjiang Wudianshi, Kuanzhai Alley, and/or Nanluogu alley in the past year. (2) The samples covered different gender and occupational groups. Considering that the customer group of cultural creative tourism is mainly youth, youth were mainly selected as interviewees, but other age groups were also taken into account. (3) In addition to ordinary tourists, we chose to pay attention to

related researchers and tourism destination managers who were more familiar with cultural creative tourism, since they understood the tourism industry from multiple perspectives.

The survey was divided into three phases, from March 2021 to July 2021. In the first stage, we went to three cultural creative tourism destinations and used intercepted survey methods to invite suitable interviewees. In the second stage, interviewees were determined through the recommendation of interviewees in the early stage. In the third stage, douban.com was used to recruit suitable interviewees to make the samples more diverse. We carefully prepared a cultural creative souvenir for each interviewee as a reward. Due to geographical constraints, online and offline interviews were conducted at the same time. The interview time for each person was controlled at 40–60 min. During the interviews, we interacted with the respondents, and confirmed and questioned their answers many times. The whole process was recorded with the knowledge and consent of the respondents. To ensure content accuracy, the sorted interview records were sent to the respondents by e-mail for confirmation within two days after their interview. The data were sorted and analyzed in time after each in-depth interview. Based on the original data, the theoretical hypothesis was constructed, preliminarily. Then, we performed analysis and correction repeatedly until new concept categories could not be extracted. This meant that the theoretical construction was saturated, and no more interviews were needed [67]. When the 24th tourist was interviewed, it had reached the saturation state in theory. After sorting, the final recording of the interview took about 15 h, with more than 62,000 words. Finally, 24 tourists were interviewed (numbers for A1–A24), 12 women and 12 men. The primarily represented age group was youth, and occupations were diversified, including students, soldiers, company employees, and government employees, as shown in Table 1.

**Table 1.** Interviewee characteristics.

| Number | Gender | Age | Occupation | Cultural and Creative Tourism Destination |
|--------|--------|-----|------------|-------------------------------------------|
| 1 | Female | 28 | Student | Wudianshi |
| 2 | Female | 30 | Student | Wudianshi |
| 3 | Female | 25 | Student | Wudianshi |
| 4 | Male | 63 | Retired professor | Wudianshi |
| 5 | Female | 21 | Student | Wudianshi |
| 6 | Male | 43 | Company clerk | Wudianshi |
| 7 | Male | 29 | Civil servant | Wudianshi |
| 8 | Male | 34 | Company clerk | Wudianshi |
| 9 | Male | 45 | Teacher | Wudianshi |
| 10 | Female | 28 | Student | Kuanzhai Alley |
| 11 | Female | 27 | Student | Kuanzhai Alley |
| 12 | Female | 25 | Company clerk | Kuanzhai Alley |
| 13 | Male | 26 | Student | Kuanzhai Alley |
| 14 | Female | 32 | Company clerk | Kuanzhai Alley |
| 15 | Female | 24 | Soldier | Kuanzhai Alley |
| 16 | Female | 18 | Student | Kuanzhai Alley |
| 17 | Male | 26 | Civil servant | Nanluoguxiang |
| 18 | Male | 26 | Programmer | Nanluoguxiang |
| 19 | Female | 26 | Sales staff | Nanluoguxiang |
| 20 | Male | 25 | Company clerk | Nanluoguxiang |
| 21 | Male | 26 | Company clerk | Nanluoguxiang |
| 22 | Female | 25 | Secretary | Nanluoguxiang |
| 23 | Male | 27 | Programmer | Nanluoguxiang |
| 24 | Male | 27 | Shopping guide | Nanluoguxiang |

In order to enhance data reliability and avoid the problem of single channel data source [68], we extracted online data with the help of web crawler technology, such as travel comments and travel notes about Kuanzhai Alley, Nanluoguxiang, and Wudianshi in the Sina Weibo, the Hornet's Nest website, the Ctrip tourism, and other websites,

published from 2020 to July 2021. The texts were cleaned and classified manually. To make the research data reliable, we strictly screened the network text during the collection process. First, the contents of news and travel guides created by tourism destinations for publicity were screened out; second, we sifted out the pure introduction and copy-paste text contents; finally, we screened out text contents with only a few words to analyze the tourist experience. Based on the steps above, 137 travel reviews and travel notes were collected, with a total of more than 52,000 words, including the title, review content, review time, etc.

*3.3. Data Analysis*

3.3.1. Open Coding

Open coding was used to extract, compare, and simplify the original data to obtain concepts and categories [69]. First, materials were elevated to concepts, which were mainly derived from original words, researcher experience, or literature, and could be words, phrases, and sentences. Second, the concepts that summarized the same or similar phenomena were gathered together and unified under the corresponding category, which was a deep-level abstraction and promotion based on conceptualization. Based on the final 190 standardization concepts, we summarized, sublimated, and compared again to find the generic relationship among concepts, then named and developed them. Finally, 42 subjective categories were extracted. The specific process is shown in Table 2.

**Table 2.** Examples of open coding process.

| Original Representative Statement in the Interview Text | Conceptualization | Categorization |
|---|---|---|
| There are folk singers in Kuanzhai Alley. They sang as they played drums. I have listened to folk songs for a long time. | Listen to folk songs | Sensory participation |
| Ear-picking is a characteristic of Chengdu. I gave it a try. | Ear-picking | |
| We can drink tea outdoors in Wudianshi. | Drink tea | |
| We run out of wine, and we can break the bowl. | Smashing the wine bowl | Physical participation |
| We made our own lipstick here. | Handmade lipstick | |
| You can feel a kind of artistic conception and specifically feel the traditional ethos of the old alley. | Traditional atmosphere | Environmental atmosphere |
| The traditional hutongs are full of the atmosphere of life, which is more natural and comfortable than the main streets. | Life atmosphere | |
| I saw a snuff bottle, which is an intangible heritage. It was very strange because one person said he was the only heir, but many stores said they were the heir. | Be strange | Thinking |
| When I saw the ancestral hall there, I thought that every village in our hometown has a family ancestral hall. | Think of hometown | Association |
| My brother and I drank and talked in the wide alley. | Drinking and talking with a brother | Tourists interact with their peers |
| I mispronounced the name of a shop there, and the locals helped me correct it. | Communication with locals | Tourists interact with local residents |
| A cyclist passed by, and I helped him take a picture. | Help others take photos | Tourists interact with other tourists |
| The architecture is very distinctive, and the house is very nice, which makes me happy. | Happy | Pleasure |
| I was excited to see these red houses at first, and I thought they were pretty good. | Excited | Novelty |

**Table 2.** *Cont.*

| Original Representative Statement in the Interview Text | Conceptualization | Categorization |
|---|---|---|
| Silversmiths can be found here. I think they are very good, and seeing them is a wonderful experience. | Wonderful | |
| The buildings there have bright colors, which attracted me at first sight. | Good-looking | Aesthetic |
| The colors blend into the local culture. It is very harmonious. | Harmonious | |

### 3.3.2. Axial Coding

Axial coding is based on open coding to discover and establish relationships among general categories and connect them by typical paradigms such as phenomenon, motivation, condition, action strategy, and result [22]. We examined the relationship among 19 initial categories obtained by open coding, and found the evidence chain; six main categories were obtained finally. The specific process is shown in Table 3.

**Table 3.** Axial coding process.

| Category | Principal Category |
|---|---|
| Sensory participation, Physical participation, Environmental atmosphere | Embodied engagement |
| Association, Imagination, Thinking | Thinking engagement |
| Tourists interact with their peers, Tourists interact with local residents, Tourists interact with other tourists | Visitor interaction |
| Cultural, Unique, Learning | Cognitive evaluation |
| Pleasure, Aesthetic, Novelty, Nostalgia | emotion |
| Episodic, Persistent, Dynamic | Memorable tourism experience |

### 3.3.3. Selective Coding

Selective coding is a cyclic process. The purpose is to select a core category from all the discovered main categories. The dominance of the core category must be proven serval times and be able to contain the most research results within a relatively broad theoretical scope, in comparison with other categories [70]. This study combined research questions and related research results, analyzed the original data, concepts, categories, and main categories, and carried out continuous comparison and repeated demonstration. We argued that the six-axis categories were dominated by the core category of "memorable tourism experience".

We selected the core category "memorable tourism experience" as the "story core" and extracted the storyline according to the collected materials; visitors received the information through physical and mental participation. The ingested information first stimulated the senses and generated sensory information registration. Meanwhile, cognitive evaluation and emotion were the processes by which subjects encoded information. After being encoded, the experience information was stored as a certain representation to form memory. As an external environmental variable, visitor interaction also influenced the whole process of forming a memorable tourism experience.

### 3.3.4. Coding Reliability

The reliability of coding mainly refers to the reliability, consistency, and stability of coding results. First, the data collection process was explained above, and was designed to guarantee data quality [68]. Second, to analyze the problem comprehensively, two graduate students participated in the coding under the guidance of two graduate tutors [11]. Finally, in terms of coding details, the research group deliberated sentence-by-sentence and discussed details repeatedly until a unified view was formed [69].

3.3.5. Theoretical Saturation Test

Theoretical sampling is one of the most common methods used to test theoretical saturation in grounded theory research. We first randomly selected one-third of the initial sample from the data collected through the network as the original data. Then, we classified the main categories derived from grounded theory as the dimensions of content analysis. No new important categories and theoretical relationships were found through the repeated induction and analysis of the new materials. Therefore, the model was considered to have reached a stable state of saturation, in theory.

ROST software was also used to count and classify the high-frequency words. We compared those results with the main category code of the model obtained from grounded theory research [71]. Moreover, the text data failed to extract new concepts and categories, and the model was highly explanatory.

**4. Results**

*4.1. Characteristics of Memorable Tourism Experiences*

1. Episodic. Memorable tourism experiences are a kind of episodic memory, i.e., the memorization, maintenance, and reappearance of a series of events related to a certain time, place, and specific situation [72]. Memorable tourism experiences include embodied and thinking engagements, visitor interaction, cognitive evaluation, and emotion. For example, when asked to recall an impressive experience, many visitors tell plots or stories: "*It was a really interesting trip. We went to taste the food, and some local specialties were delicious and interesting*" (A11).

2. Persistent. The self-citation effect suggests that we seem to remember personal information better than nonpersonal information because we are particularly interested in it [73]. Memory caused by important, dramatic, and surprising public events is called flashbulb memory [74]. This phenomenon is important in the formation of memorable tourism experiences, which are inseparable from personal experiences. Moreover, memorable tourism experiences are often related to individual participation and emotion, and a certain importance and novelty. Therefore, although tourists only experience the process of traveling once, profoundness is still a characteristic of memorable tourism experiences: "*I fell in love with these buildings at first sight, and they were so colorful that I still remember them vividly*" (A3).

3. Dynamic. Memorable tourism experiences also have the characteristic of dynamic development, which is mainly reflected in two aspects: memory bias and memory awakening [75]. Interference theory points out that forgetting in LTM does not occur because of the passage of time, but because other memories interfere with the extraction of the information one is trying to recall, especially when other memories are similar to the memory one wants to recall. The repeated experiences in cultural creative tourism and external information can cause memory bias. "*At first, I thought the souvenirs were good, but later I went to many places and found that they were all the same*" (A6). The memorization consists of three links: retention, reproduction, and recognition. After visiting scenic spots, tourists will still be affected by some off-site texts, especially extended ones. This leads to the reproduction of tourism experience memory, which is also the process of memory awakening. Memory awakening is helpful for tourists to recall their travel experiences and stimulate their emotions: "*when the photos were developed, it occurred to me that some of the scenery and decorations there were very nice*" (A10). Moreover, it is an important driving force to promote tourists' recommendations and revisit behaviors. Due to the multifaceted and complex nature of experience memory, the experience memory evoked by different clues and at different times also has certain differences.

*4.2. Influencing Factors of Memorable Tourism Experiences*

Tourist engagement is a process of information intake, and the degree to which tourists devote themselves to service production and delivery is mainly demonstrated through a high degree of interaction and collaboration [9,76]. This interaction influences the genera-

tion of memorable tourism experiences through the mediating effect of cognitive evaluation and emotion. When talking about memorable experiences, most tourists mentioned the process of experiencing the local culture through embodied and thinking engagements deeply: "*What impressed me most was eating local specialties, which we cannot get in our area*" (A8). These results suggested that embodied and thinking engagements were important antecedents to the formation of memorable tourism experiences. Embodied engagement is also an important factor in stimulating thinking engagement.

Embodied engagement. Embodied cognition theory holds that human cognition is deeply rooted in the interaction between the body and the world, and that embodied engagement mainly achieves the purpose of experience through the contact between body organs and landscape resources [54]. Embodied engagement is composed of sensory partici­pation, physical participation, and environmental atmosphere. These three categories are in line with the characteristics of cultural creative tourism. Embodied engagement is not only limited to sensory participation, such as tasting food and appreciating architecture but also includes deep-seated physical participation in tourism activities, such as making artwork and souvenirs, participating in games, and other ways. The environmental atmosphere is also very important content, such as simple streets and lively atmosphere: "*The original hutongs made me feel like I had traveled to ancient times*" (A15).

Thinking engagement. In addition to embodied engagement, thinking engagement is also an important way for tourists to obtain experience. When tourists see a landscape with special significance, they generate new associations and imagination according to their own experiences and memories and extend their thoughts to different situations, thus affecting their cognitive evaluation and emotion [77]. Thinking engagement mainly includes imagination, thinking, and association. Imagination is a psychological process of forming a new image through processing and transformation across time and space on the basis of a specific landscape and situation: "*When I see many Zhuangyuan tablets, I think this family used to be so big*". Thinking is an exploratory activity based on questions with certain initiative and creativity: "*I saw an intangible cultural heritage called a snuff bottle, which is quite interesting. At that time, I was very surprised that many shops said they were the only inheritor*" (A18). Association is usually an event related to the subject's own experiences caused by a specific landscape and situation, it is often accompanied by contrast: "*When I see ceramics, I immediately think of ceramics made in Japan. I think Chinese ceramics are not inferior to those made in Japan because Japan also learned from China*" (A1).

Tourist interaction. Tourist interaction refers to the communication and interaction between tourists and other tourists or local residents [78]. It can not only enhance mutual understanding and emotion transfer among people—"*The children here are very happy. I also feel very happy*" (A12)—but also improve the memorable tourism experiences—"*I remember talking to a resident about some of the cultural customs*" (A4). Tourist interaction is the external influence factor, it plays a regulating role in embodied and thinking engagements on cognitive evaluation and emotion—"*I took a rickshaw pulled by the local people, and he talked to me a lot. I felt that I had a deeper understanding of the local culture*" (A20).

Cognitive evaluation. After registering the stimulus information in the sensory system through engagement and interaction, tourists judge and evaluate it with a certain think­ing mode [79]. Such original information is represented in a certain form, thus forming memorable tourism experiences. Subsequently, cognitive evaluation further affects tourist emotions: "*I think Sichuan opera is unique there. I've never seen it before. I was very happy to see it today*" (A6). The cognitive evaluation formed in cultural creative tourism mainly includes uniqueness, culture, and learning. Uniqueness is a distinct personality that distinguishes it from other landscapes. Tourists realize the uniqueness of the landscape through asso­ciation and imagination: "*When I saw that there were many top scholars in the ancestral hall, I thought that there were also family ancestral halls in my hometown*" (A16). Culture refers to the landscape, related to local culture, architecture, and customs, which can be considered the soul of a place: *There is a Linjia restaurant, which is deeply impressed by the southern Fujian cuisine. The tables and chairs in the restaurant and the partition are all of the southern Fujian*

*styles*" (A17). Learning means that the landscape resources of a cultural creative tourism destination have certain connotations, which can provide a way for people to explore new knowledge and improve themselves: "*There is a mural that is quite impressive. I can not only take photos but also know what the life of people in Chengdu was like before and understand some difficult processes*" (A7).

Emotion. As the experience of people's attitudes toward things, emotion is a reflection of whether their needs are satisfied. Therefore, emotion is also one of the important factors affecting the formation of memorable tourism experiences. This kind of memory is called emotional memory, which has the characteristics of vividness and profoundness, and, situationally, it is often more solid than other memories. Sometimes, the facts of an experience are forgotten, but the excitement or frustration remains in the memory [80]. The emotional experience memories presented by the interview samples can be summarized into four aspects: pleasure, aesthetic, nostalgia, and novelty. The pursuit of pleasure is the fundamental and ultimate goal for tourists to travel. The pleasure of tourists comes from many aspects, including relaxation away from daily hassles and the happiness generated by participating in tourism projects: "*I like the slightly bright colors of Wudianshi architecture, which makes me feel happy*" (A5). Aesthetic is the subjective feeling of a subject to objective reality, which has a level difference. It can represent the color that makes an impression: "*The buildings in Wudianshi are very beautiful. I like colorful things very much*" (A5). Artistic beauty is also inherent in architectural design: "*The architectural culture of Wudianshi in southern Fujian is unique and exquisite in material, plane, and structure, which has certain aesthetic value*" (A22). Nostalgia is the feeling of remembering the past. Old things, old people, hometowns, and lost years are the most common subjects of nostalgia. In scenic spots, it is generally an emotion generated by experiencing things related to our past experiences: "*I remember those tapes and old CDS, and when I look at those things, I realize I have heard a lot of them back then*" (A2). The sense of novelty generally comes from the local unique or special connotation of landscape resources, often accompanied by unexpected surprises and excitement: "*There is a special snack called three cannons—three balls rolled in sesame seeds and then hit a gong; this process is fun*" (A13).

### 4.3. Formation Mechanism of Memorable Tourism Experiences

The core of forming memorable tourism experiences in cultural creative tourism is the on-site tourist experiences. As shown in Figure 1, from the perspective of embodied cognition, this study deduced that embodied and thinking engagements were two important ways for tourists to interact with tourist attractions and absorb information in cultural creative tourism. Both influenced memorable tourism experiences through the mediating effects of cognitive evaluation and emotion. Therefore, embodied and thinking engagements were defined as antecedent factors of cognitive evaluation and emotion in this research. Tourism interaction was defined as an external situational factor which had a moderating effect in the transformation from embodied engagement and thinking engagement to cognitive evaluation and emotion. The cognitive evaluation theory of emotion generation holds that emotion generation is the psychological result of a subject's cognitive evaluation of an object and its environment. Learning, culture, and uniqueness, identified by interviews and coding, could promote the generation of tourists' sense of pleasure, aesthetic, nostalgia, and novelty. As an antecedent element of emotion, cognitive evaluation acts on memorable tourism experiences through emotion. However, memorable tourism experiences are also dynamic, affected by some subjective and objective factors that produce memory bias phenomena. Stored memories can be extracted and recognized through off-site texts, and this extraction process is the memory awakening process.

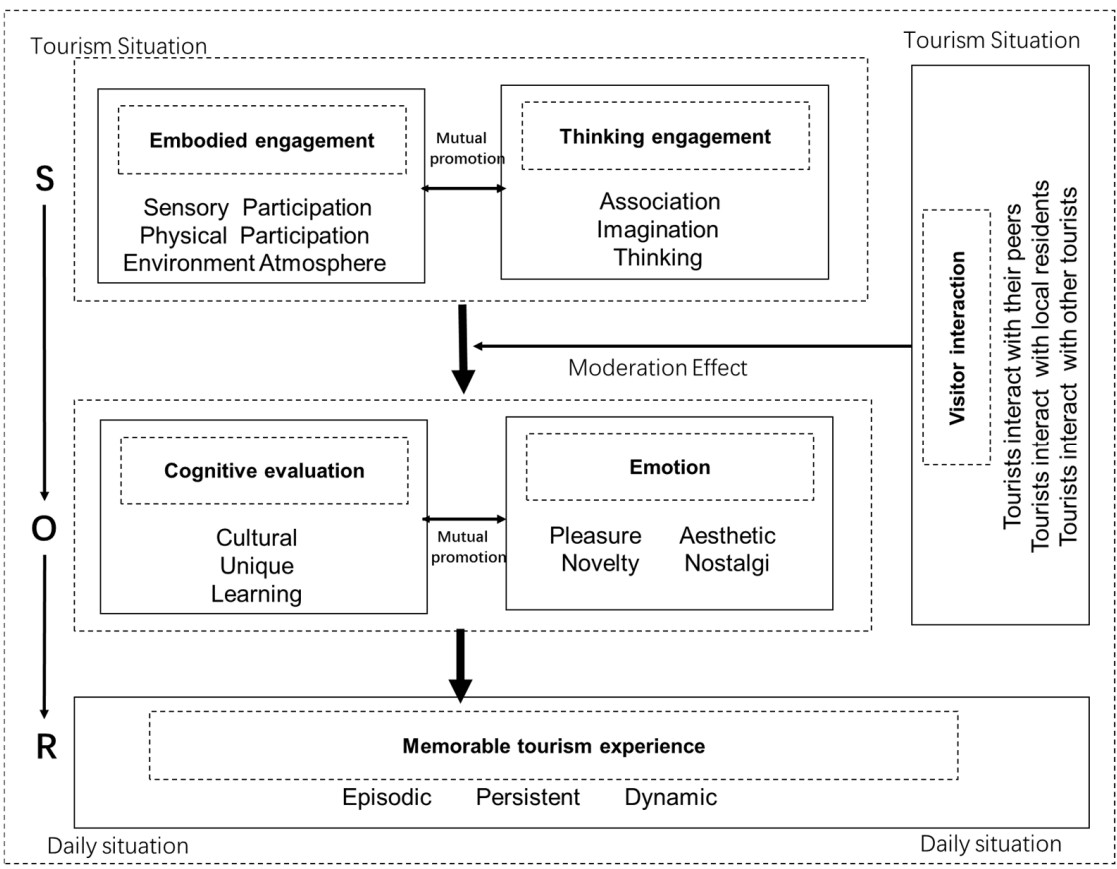

**Figure 1.** Model diagram of memorable tourism experience.

## 5. Conclusions and Discussion

### 5.1. Conclusions

To further understand and explore the concept, connotation, influencing factors, and relationship of memorable tourism experiences, this study used qualitative research methods to explore new understanding and recognition of memorable tourism experiences in cultural creative tourism. The conclusions were as follows:

First, the concept and characteristics of memorable tourism experiences were discussed. The memorable tourism experience is a kind of episodic memory, which is a self-related autobiographical memory, including image and emotional memories. It also presents episodic, persistent, and dynamic characteristics. This was consistent with Kim et al. (2012) and Tung et al.'s (2011) study results that travel experience memory was an autobiographical memory and the essence of memorable tourism experiences was storytelling [6,19]. In this study, the connotation and characteristics of memorable tourism experiences were systematically sorted out. The dynamic characteristics of memorable tourism experiences were summarized.

Second, the influencing factors of memorable tourism experiences in cultural creative tourism were identified, including embodied engagement, thinking engagement, visitor interaction, cognitive evaluation, and emotion. Embodied engagement mainly included sensory participation, physical participation, and environmental atmosphere. Thinking engagement mainly included association, imagination, and thinking. Previous studies pointed out that cultural contact was an important way to improve tourism experiences in cultural tourism, and cultural contact was mainly realized through tourist engagement [9]. However, this study found that embodied engagement and thinking engagement in cultural and creative tourism were the basis for the formation of memorable tourism experiences. Visitor interaction mainly included tourists interacting with their peers, tourists interacting with local residents, and tourists interacting with other tourists. Tourist interaction can

promote emotional transmission and enhance the emotional connection between people. Many studies showed that tourist interaction was an important factor to improve tourism experiences [78]. Cognitive evaluation mainly included cultural, unique, and learning experiences. Emotion mainly included pleasure, novelty, aesthetics, and nostalgia. These factors were partially consistent with the dimension composition of memorable tourism experiences proposed by Kim et al., (2012) [19], but this study also found some special dimensions of memorable experiences based on cultural creative tourism background, such as nostalgia.

Last, the formation mechanism model of memorable tourism experiences was constructed. The core process of the formation of memorable tourism experiences was the process of the on-site tourist experience [19]. Embodied and thinking engagements were the antecedent variables of cognitive evaluation and emotion, both of which influenced memorable tourism experiences through the mediating effects of cognitive evaluation and emotion. Further, cognitive evaluation influenced memorable tourism experiences through the mediating effect of emotion. Visitor interaction played a regulating role in embodied and thinking engagements, as well as cognitive evaluations and emotional relationships.

### 5.2. Theoretical Contributions

First, existing literature has focused on the influence of memorable tourism experiences on tourists' behavioral intentions [9,49–51]. Most studies were carried out from the perspective of current experiences, but lack of exploration on the changing process and formation mechanism of memorable tourism experiences. In view of the research gap mentioned above, this study began from embodied cognition theory and constructed the formation mechanism model of memorable tourism experiences in cultural creative tourism by identifying structural dimensions. The theoretical model showed the relationship between various factors and the roles they played in the formation of memorable tourism experiences clearly. It also emphasized the important role of embodied engagement in memorable tourism experiences [54]. This study expanded and enriched the research scope and framework of memorable tourism experiences. It provided a theoretical basis for empirical research and practical management of memorable tourism experiences in cultural creative tourism.

Second, this study used interviews to collect data and identified the key dimensions of memorable tourism experiences through layer-upon-layer coding, which confirmed the planned learning theory and cognitive evaluation theory of emotion. On the basis of the technical advantages of grounded theory [64,65], we not only discovered the five elements of embodied engagement, thinking engagement, visitor interaction, cognitive evaluation, and emotion that affected memorable tourism experiences, but also extracted some new categories. The research, innovatively, divided embodied engagement into three dimensions: sensory participation, physical participation, and environmental atmosphere. Compared with other studies, this research emphasized the important role of the environmental atmosphere in embodied engagement and expanded its original connotation dimension [9,30]. Moreover, this study extracted the new category of nostalgia, which was in line with the characteristics of cultural creative tourism, and further excavated and expanded Kim's (2012) original seven-factor theory. The discovery of these dimensional categories laid a foundation for the subsequent development of measurement scales. This conclusion not only expanded the application of embodied cognition theory in tourism experience but also echoed scholars' suggestions that memorable tourism experiences could have differences in different research situations, and responded to the call for more diversified research on memorable tourism experiences [10,11].

Last, the concepts and characteristics of memorable tourism experiences were further explored. Existing literature mainly discussed the connotation and the dimension composition of memorable tourism experiences from its contents and influencing factors [6,19]. This study innovatively regarded memorable tourism experiences as a kind of episodic memory, which had episodic, persistent, and dynamic characteristics. It expanded the

connotation and research perspective of memorable tourism experiences and established a foundation for grasping its essential rule.

### 5.3. Implications for Managers

Exploring local culture and blending in creative elements, tourists travel to cultural creative tourism destinations mostly for the motive of cultural experience [9,11]. The authenticity of culture is particularly important for improving their travel experience. The higher the authenticity of culture is, the more unique it is, and the more likely it is to leave profound memories for tourists [81]. Creativity is able to produce new things, which are unique, original, and meaningful. It relies on the wisdom, skills, and talents of creative people to create and improve cultural resources and produce high-value-added products with the help of high technology, which is conducive to activating cultural heritage and enhancing the knowledge and diversity of tourism projects [82].

Highlighting participation, mobilizing visitors' senses and thinking fully, tourism participation is a means for tourists to obtain information and important content of tourism experience and memory [9,81]. First of all, managers should pay attention to the application of new technologies, such as modern VR and AR, to display the local folk culture and historical features authentically by reproducing the original life pictures and scenes of historical figures. Second, the tourist destinations should create a certain atmosphere, for example, setting up a distinct theme and immersive tourism places, and enhance tourists' sense of involvement and ceremony to activate and extend tourists' five senses (vision, hearing, touch, smell, and taste) and related experiences. Finally, the destination could design interactive programs and pay attention to promoting interaction between people, such as organizing competitive competitions, interactive games, and DIY crafts.

Regarding the activation of emotions to impress tourists: emotional memory refers to the memory of emotions once experienced. Although the events that caused the emotions have passed, the emotional experiences can be relatively stable and lasting in memory [80]. Therefore, cultural creative tourism destinations should pay extra attention to spiritual connotation and atmosphere creation in product design to promote tourists' emotional experience. In particular, nostalgia has become an important part of cultural and creative tourism in recent years [83]. Therefore, the design and atmosphere creation of nostalgic products should be noticed.

Strengthening the design of representative cultural creative products could drive the local commodity economy and improve the participation and experience of tourists. Cultural and creative tourism destinations should make full use of new media, improve publicity of scenic spots, create marketing topics, and highlight participation and interaction. This could help enhance and awaken the experience memory of tourists to promote their recommendations and revisit behavior.

### 5.4. Limitations and Further Research

We conducted in-depth research on memorable tourism experiences and their formation mechanisms in cultural creative tourism; these had certain theoretical values and practical significance. However, there were also some limitations due to personal abilities and resources. For example, the applicability and extensibility of the results warrants further discussion. There was also a data acquisition limitation, and a lack of quantitative verification. The study of memorable tourism experiences is a relatively emerging research field, which must be refined and deepened in many aspects. Future studies must adopt more diversified research methods—for example, the experimental method—to further study memorable tourism experiences in different tourism situations. Secondly, the research background of memorable tourism experiences should be expanded to other cultural or natural scenic spots, and the rules and characteristics of memorable tourism experiences, such as outbound travel and self-driving travel, should be discussed further. Finally, individual differences should be considered deeply in future studies of memorable tourism experiences.

**Author Contributions:** Conceptualization, Z.Y. and A.H.; methodology, Z.Y.; software, Z.Y.; validation, Z.Y. and A.H.; formal analysis, Z.Y.; investigation, Z.Y. and J.W. resources, A.H.; data curation, Z.Y.; writing—original draft preparation, Z.Y. and A.H.; writing—review and editing, Z.Y., J.W. and A.H.; visualization, Z.Y., J.W. and A.H.; supervision, A.H.; project administration, A.H.; funding acquisition, A.H. All authors have read and agreed to the published version of the manuscript.

**Funding:** This research was funded by [Study on Gene decoding and cultural pedigree Construction of Fujian "Fu Culture"] grant number [FJ2022TWF002] And the APC was funded by [Anmin Huang].

**Informed Consent Statement:** Informed consent was obtained from all subjects involved in the study.

**Data Availability Statement:** The data presented in this study are available on request from the corresponding author. The data are not publicly available due to privacy restrictions.

**Conflicts of Interest:** The authors declare no conflict of interest.

## Appendix A

Outline of the open interview on memorable tourism experience in cultural creative tourism.

1. Preparation stage

    ◇ Brief introduction of the background, purpose and significance of the interview
    ◇ The researchers emphasize that it is for research purposes only, and promise to keep the interview results confidential
    ◇ Simple, small talk can be used to understand the personal information of the interviewees and remove the interviewees' psychological guard, and gradually talk about their real thoughts and feelings.

2. Expectations before travel

    ◇ Why do you choose to visit this cultural creative tourism destinations?
    ◇ What do you think of this cultural creative tourism destinations before you travel? What are your expectations?

3. Travel experience

    ◇ Could you give a brief description of your visit?
    ◇ What impressed you most during your visit to the cultural creative tourism destinations?
    ◇ What were your favorite and least favorite aspects of the tour?

4. Evaluation after travel

    ◇ How does your actual experience compare to your expectations?
    ◇ Do you casually recall the trip?
    ◇ How has your evaluation and attitude of that trip changed compared to the on-site experience?
    ◇ What factors do you consider when choosing a cultural creative tourism destinations once again?

5. Appreciate

    Thanks for your support!

6. Research Background (please tick "√" before the item that fits your situation):
    1. Your gender: ☐ Male ☐ Female
    2. Your Age: ☐ Below 20 years ☐ 20–25 years ☐ 26–35 years ☐ 36–45 years ☐ 46–59 years ☐ 60 years and above
    3. Your occupation is _______.
    4. Which cultural creative tourism destinations have you visited: ☐ Jinjiang ☐ Wudianshi ☐ Nanluoguxiang ☐ Kuanzhai alley.

    Thank you again for your warm help and strong support! All the best!

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
