# Peer review of "Memorable Tourism Experiences’ Formation Mechanism in Cultural Creative Tourism: From the Perspective of Embodied Cognition"

_sustainability, doi:10.3390/su15054055_

Round 1

Reviewer 1 Report

I would like to congratulate the authors for the paper as I find it any interesting.

Please see the following suggestions for improving the work.

General observations

The entire paper has to be revised in terms of English because there are phrases that are confusing, singular and plural issues and many grammatical and punctuation issues.

Introduction

The introduction is well developed as it introduces the reader to the problem at hand and contextualizes what is important according to the authors. Among these ideas is the increasing interest in the embodiment dimensions paradigm in tourism studies and memorable experiences.

Suggestions for improvement

Please see the examples to correct

Although most scholars agree that satisfaction is the main index to predict tourists’ intention to revisit, some scholars believe that compared with satisfaction, memory (as a sublimation of the tourism experience) is a more accurate indicator to predict it [3].

Although I agree with the sentence when I see “some authors”… I want to see more examples. So I suggest you add more examples and more recent ones.

Line 32 The field of research on MTEs need needs to be further expanded.

Line 43 Cultural creative tourism as an emerging tourism form of tourism that integrates the

It is already a more consolidated form of tourism so please rephrase the emerging

Cultural creative tourists are active and interested, physical participation and involvement is an important way to optimize cultural creative experience [26].

Please revise the English in this sentence

Previous studies have shown the positive impact of sensory experience on memorable tourism experiences [8]. Line 61

Please change to experiences

Line 74 The key point of this study is to describe the tourism experience from the memory perspective, and the MTEs theory will be introduced into the situation of cultural creative

Change Will be added to the context of cultural creative tourism

Literature review

Cultural Creative tourism

This section is very poor as creative tourism literature is badly presented and revised here. Major authors must be included here as the evolution of creative tourism is very poorly treated here. The connection with the ICCs (cultural and creative industries) is  only one of several approaches to the development of creative tourism models and evolution in its consumption. So several recent authors must be included here such as

Carvalho, R., Costa, C., & Ferreira, A. M. (2019). Review of the theoretical underpinnings in the creative tourism research field. Tourism & Management Studies, 15(SI), 11–22. https://doi.org/10.18089/tms.2019.15si02

Carvalho, R., & Reis, P. (2022). Surfing the creative wave: Designing surfing as a creative tourism experience. In R. Augusto Costa, F. Brandão, Z. Breda, & C. Costa (Eds.), Planning and Managing the Experience Economy in Tourism (pp. 189–214). IGI Global. https://doi.org/10.4018/978-1-7998-8775-1

Richards, G., & Duxbury, N. (2021). Trajectories and trends in creative tourism: Where are we headed. In N. Duxbury, C. P. Carvalho, & S. Albino (Eds.), Creative Tourism: Activating Cultural Resources and Engaging Creative Travellers (pp. 53–58). CABI. https://doi.org/10.1079/9781789243536.0007

Authors need to approach the development of tourism consumption imposed by the creative tourism paradigm, emphasizing the co-creation, participation and creativity as a whole in the production and consumption of cultural and creative tourism experiences. Not only the ICCs mentioned before.

Memorable tourism experiences

Here the first three paragraphs are very confusing as memorable experiences are poorly introduced in the context of tourism studies. First authors should introduce the concept, and then talk about the scales. And key studies of this connection must be applied when relating memorable tourism experiences with creative and cultural tourism For instance:

Seyfi, S., Hall, C. M., & Rasoolimanesh, S. M. (2019). Exploring memorable cultural tourism experiences. Journal of Heritage Tourism, 6631(15), 3. https://doi.org/10.1080/1743873X.2019.1639717

This connection must be attained here.

Embodied cognitive theory

In this section, it is clear what the authors want to convey. Consider mentioning briefly, Merleau Ponty´s embodied subjectivity theory here. It can add value to the paper.

Grounded theory and content analysis method

This section cannot be here in the literature review section as it is an important requisite of the methodology so needs to go into the next section.

Method

I liked this section as detail was a major concern in this section. Coding procedures, sum up tables as it is needed in qualitative research. So on this matter, it is well written. Congratulations. Please add a citation author in the coding reliability sub-section to further reinforce this section.

Results

I recommend that transcriptions of interviews should be in italic.

Conclusions

 No problems here

Reviewer 2 Report

Dear authors,

Thank you for this interesting research contribution in topic of memorable tourism experiences from the perspective of embodied cognition. 

After carefully reading your paper I must say I am really impressed with the quality of work you did. You cited the most prominent literature and conducted a detailed analysis of embodied cognition theory.

My minor suggestion would be to add at least three further research direction and elaborated each in couple of sentences, since readers would expect this from your contribution.

Congratulations once again and good luck with publishing your paper.

Reviewer 3 Report

Thank you for letting me review this manuscript with an interesting topic. The following comments may help the author further improve the quality of the paper.

1. I Believe the Intro should be improved by better articulating the significance and contribution of the study. It is not enough to only highlight the gap but it is important to show why addressing this gap is of significant importance. 
2. The theoretical underpinning is well developed. However, some claims should be supported. Still, some important references are missing, such as https://doi.org/10.1177/0047287522113805, https://doi.org/10.1016/j.tourman.2022.104527; https://doi.org/10.1016/j.tourman.2019.104006

3. You need to justify how to ensure the trustworthiness of your data from the participatory observation and the interviews.
4. How did you design the interview questions? How was the data collected? Lot of information was missing.

5. Theoretical contributions and the discussion with references to literature were very limited. The practical implications were also very brief.

6. The work should be proofread to improve the standard of English.

Round 2

Reviewer 1 Report

The submission was improved and i recommend its publication

Reviewer 3 Report

The quality of the manuscript has greatly improved as a result of the revisions!
